# Thrombosis and Hyperinflammation in COVID-19 Acute Phase Are Related to Anti-Phosphatidylserine and Anti-Phosphatidylinositol Antibody Positivity

**DOI:** 10.3390/biomedicines11082301

**Published:** 2023-08-18

**Authors:** Jaume Alijotas-Reig, Ariadna Anunciación-Llunell, Stephanie Morales-Pérez, Jaume Trapé, Enrique Esteve-Valverde, Francesc Miro-Mur

**Affiliations:** 1Systemic Autoimmune Diseases Research Unit, Vall d’Hebron Institut de Recerca (VHIR), 08035 Barcelona, Catalonia, Spain; ariadna.anunciacion@vhir.org; 2Systemic Autoimmune Diseases Unit, Department of Internal Medicine, Hospital Universitari Vall d’Hebron (HUVH), 08035 Barcelona, Catalonia, Spain; 3Department of Medicine, Faculty of Medicine, Universitat Autònoma de Barcelona (UAB), 08035 Barcelona, Catalonia, Spain; 4Systemic Autoimmune Disease Unit, Internal Medicine Department, Althaia Healthcare University Network of Manresa, 08243 Manresa, Catalonia, Spainjtrape@althaia.cat (J.T.); 5Systemic Autoimmune Diseases Unit, Department of Internal Medicine, Hospital Universitari Parc Taulí, 08208 Sabadell, Catalonia, Spain

**Keywords:** antiphospholipid antibodies, anti-phosphatidylserine antibody, anti-phosphatidylinositol antibody, COVID-19, disease severity, thrombosis, long COVID-19

## Abstract

Antiphospholipid antibodies (APLA) are strongly associated with thrombosis seen in patients with antiphospholipid syndrome. In COVID-19, thrombosis has been observed as one of the main comorbidities. In patients hospitalised for COVID-19, we want to check whether APLA positivity is associated with COVID-19-related thrombosis, inflammation, severity of disease, or long COVID-19. We enrolled 92 hospitalised patients with COVID-19 between March and April 2020 who were tested for 18 different APLAs (IgG and IgM) with a single line-immunoassay test. A total of 30 healthy blood donors were used to set the cut-off for each APLA positivity. Of the 92 COVID-19 inpatients, 30 (32.61%; 95% CI [23.41–43.29]) tested positive for APLA, of whom 10 (33.3%; 95% CI [17.94–52.86]) had more than one APLA positivity. Anti-phosphatidylserine IgM positivity was described in 5.4% of inpatients (*n* = 5) and was associated with the occurrence of COVID-19-related thrombosis (*p* = 0.046). Anti-cardiolipin IgM positivity was the most prevalent among the inpatients (*n* = 12, 13.0%) and was associated with a recorded thrombosis in their clinical history (*p* = 0.044); however, its positivity was not associated with the occurrence of thrombosis during their hospitalisation for COVID-19. Anti-phosphatidylinositol IgM positivity, with a prevalence of 5.4% (*n* = 5), was associated with higher levels of interleukin (IL)-6 (*p* = 0.007) and ferritin (*p* = 0.034). Neither of these APLA positivities was a risk factor for COVID-19 severity or a predictive marker for long COVID-19. In conclusion, almost a third of COVID-19 inpatients tested positive for at least one APLA. Anti-phosphatidylserine positivity in IgM class was associated with thrombosis, and anti-phosphatidylinositol positivity in IgM class was associated with inflammation, as noticed by elevated levels of IL-6. Thus, testing for non-criteria APLA to assess the risk of clinical complications in hospitalised COVID-19 patients might be beneficial. However, they were not related to disease severity or long COVID-19.

## 1. Introduction

Since the description of the first cases of patients infected with the severe acute respiratory distress syndrome coronavirus-2 (SARS-CoV-2) virus with severe clinical forms of coronavirus disease 2019 (COVID-19) [1,2,3,4], there has been increasing evidence pointing to a coagulopathy based on various hemostatic laboratory parameters [5,6,7]. Microthrombosis, thrombosis, and thromboembolic events have been described in fatal cases [8,9,10,11,12,13]. In addition, the relationship between inflammatory and clot pathways seems to play an important role in the pathophysiology of severe forms of COVID-19 [14,15]. When tissues are damaged or infected, inflammatory responses are triggered, leading to the release of various signalling molecules, such as cytokines and chemokines. These signalling molecules not only recruit immune cells to the site of injury but also activate the complement and clotting pathways [12,13]. Inflammatory cells, particularly neutrophils and monocytes, release procoagulant substances, for instance interleukin-1 or tissue factor, and serine proteases such as elastase and cathepsin G, that enhance clot formation [16,17,18]. Furthermore, inflammatory molecules can activate platelets and promote their aggregation [19,20]. In turn, clotting factors, namely thrombin, can directly activate immune cells and stimulate the release of inflammatory mediators [21].

Antiphospholipid antibodies (APLA) are a heterogeneous group of antibodies against different phospholipids, including anti-phosphatidylserine (aPS), anti-phosphatidylinositol (aPI), anti-phosphatidylglycerol (aPG), anti-cardiolipin (aCL), anti-phosphatidylethanolamine (aPE), or anti-phosphatidic acid (aPA). Other APLA are directed against proteins that form complexes with phospholipids, such as anti-β2-Glycoprotein I (aβ2GPI), anti-prothrombin (aPT) or anti-annexin V (aAnV) [22]. APLA might exert its pathogenic role through increased activation of the complement and clot pathways [23,24]. Some of these APLA (aCL and aβ2GPI), known as criteria APLA, underlie the pathogenesis of antiphospholipid syndrome (APS), clinically defined by the presence of arterial or venous thrombotic events or pregnancy morbidity [25].

APLA can also be produced in response to infections [26,27]. Viral infections induce APLA by molecular mimicry mechanisms, although its thrombogenic role is not well defined [28]. However, coagulopathy related to COVID-19 was associated with the presence of APLA [29]. In addition, APLA with functional prothrombotic activity was detected in the sera of hospitalised patients with COVID-19 [30], and their presence correlated with clinically severe COVID-19, suggesting a causal role for APLA in the coagulopathy related to COVID-19 [31]. Studies reporting the prevalence of APLA in patients with COVID-19 focused mainly on criteria APLA [32,33], with only a few delving into non-criteria APLA, mainly aPS or aPT [34,35]. Therefore, the prevalence of non-criteria APLA is relatively unknown. Furthermore, the clinical value of APLA positivity in these patients is a matter of debate, as are their titres, since the percentage of recurrent positive APLA tests is low [36,37,38].

The aim of our study was to assess the relationship between positivity of different APLAs, including criteria and non-criteria APLAs, during the acute phase of disease with thrombosis, inflammation, or clinical parameters related to the severity of COVID-19. Eventually, we verified its association with post-acute disease outcomes, such as long COVID-19.

## 2. Materials and Methods

### 2.1. Patients

A total of 92 patients with PCR-documented SARS-CoV-2 infection and clinical data compatible with COVID-19 aged ≥18 were included in this study. All included cases were recruited from hospitalised patients with COVID-19 during the first outbreak of COVID-19 (March–April 2020) at two tertiary centres, the Vall d’Hebron Hospital Universitari in Barcelona (Spain) and Fundacio Althaia Xarxa Assistencial in Manresa (Spain).

Informed consent from the patient was not required due to the type of study and the emergency during the first outbreak of COVID-19. An annotation was made in the patient’s medical record. Blood samples taken for clinical management were used to assess APLA levels. Demographic, clinical, and laboratory data were collected in a database set. When the patient’s medical history indicated that the patient had suffered a thrombosis before being hospitalised for COVID-19, this was recorded as a history of thrombosis. If the thrombotic event occurred during their hospitalisation for COVID-19, it was recorded as a COVID-19-related thrombosis. A total of 83.5% of patients hospitalised for COVID-19 were under anticoagulant treatment with low molecular weight heparin: 69.2% were on weight-adjusted prophylactic doses and 14.3% on therapeutic doses.

One patient did not consent to participate in the study, and their entire data was removed from the study.

The included cases were clinically classified into 2 groups: moderate (*n* = 78, 84.8%) and severe (*n* = 14, 15.2%). Moderate include those hospitalised patients who required non-invasive oxygen support with a fraction of inspired oxygen < 0.6 (FiO_2_ < 0.6). Those inpatients with higher oxygen supply requirements were included in the severe group. Of the severe group, a total of 8 patients died during their hospitalisation.

A group of 30 healthy individuals was used as a control to establish the cut-off point for APLA positivity. These individuals were blood donors at the Blood and Tissue Bank of Catalonia (Banc de Sang i Teixits, BST, Catalonia). The serum of these patients was collected 2 years before the start of the pandemic with informed consent approved by the Vall d’Hebron Hospital Universitari ethical committee (PR(AMI)197/014). The frozen sera were selected from HC who matched the sex (42 (45.7%) females in COVID-19 inpatients vs. 13 (43.3%) females in HC, *p* = 0.50, Fishers’ exact test) and age (mean ± standard deviation, 63.7 ± 13.0 years in COVID-19 inpatients vs. 60.2 ± 15.4 years in HC, *p* = 0.29, Mann–Whitney U test) of our cohort of COVID-19 inpatients.

### 2.2. Laboratory Methods

Basic blood tests and standard biochemistry parameters were analysed. Acute phase reactants such as C-reactive protein (CRP), lactate dehydrogenase (LDH), fibrinogen, ferritin, procalcitonin, and IL-6 were tested. Furthermore, clot tests such as prothrombin time (PT), activated partial thromboplastin time (aPTT), platelet count, and D-dimer were also tested.

#### Multiplex Line Immune Assay (LIA) for APLA Detection

A blood sample was taken by venipuncture (Vacutainer^®^, Becton-Dickinson Co., Franklin Lakes, NJ, USA), and serum was separated after clotting by centrifugation at 1500× *g* for 15 min. The hemolytic sera were discarded, and the samples were stored in aliquots at −20 °C. Multiplex detection with a single procedure of lipid reactive IgG and IgM antibodies aCL, aPE, aPS, aPA, aPG, and aPI, as well as the phospholipid binding protein antibodies aβ2GPI, aAnV, and aPT, was performed in diluted serum samples (1:33) following the manufacturer’s recommendations (GA Generic Assays GmbH, Dahlewitz, Germany) and already described [39,40,41]. Test results were assessed using the lot-specific evaluation template provided by the manufacturer. APLA levels were considered when the respective APLA showed a stronger signal than the band in the evaluation template (1 = none, 2 = low, 3 = medium, and 4 = high). In parallel, the cohort of 30 healthy blood donors (HC) was used to establish the cut-off for the determination of APLA positivity. Sera from HC reached maximum APLA levels of 1 (none) except for aCL IgG, aPG IgG, and aPI IgG, which reached APLA levels of 2 (low) (Figure 1). Thus, these three APLA (aCL IgG, aPG IgG, and aPI IgG) were deemed positive when their levels were ≥3, and for the rest of the APLA when their levels were ≥2.

Laboratory criteria required for classification as APS are described in Miyakis et al. [42]. It is indicated that positive tests for aCL and aβ2GPI should be performed at least twelve weeks apart. In our cohort, although 65 (70.1%) patients were tested twice, it was completed in less than twelve weeks.

### 2.3. Statistical Analysis

Data were analysed using R software (version 4.2.1, R Core Team, Vienna, Austria) and the packages dplyr, tidyverse, and R-stats. Continuous variables were summarised as mean and standard deviation (SD) or medians and interquartile ranges (IQR). Descriptive statistical analyses were performed using an unpaired *t*-test for continuous parametric variables or the Mann–Whitney U test for non-parametric variables. The assumption of normal distribution was tested with Shapiro–Wilk tests (*p* > 0.05 assuming normally distributed data). Categorical variables are presented as absolute numbers and relative frequencies. For categorical variables, Fisher’s exact test or the chi-squared test were used. A two-sided α-level of 0.05 or less was statistically significant. To explore any APLA positivity as a risk factor associated with COVID-19 severity or long COVID-19, logistic regression models were used (OR [95% CI]). Statistical analyses were carried out in the Statistics and Bioinformatics Unit (UEB-VHIR).

## 3. Results

### 3.1. Prevalence of APLA in COVID-19 Hospitalised Patients

During the first outbreak of COVID-19 in March 2020, 92 inpatients who tested positive for SARS-CoV-2 by PCR were recruited, whose demographic and previous clinical data are reported in Table 1.

A total of 298 samples from the 92 patients and sera from 30 healthy controls (HC) were analysed with a multiplex line immunoassay (LIA) that screened for eighteen different APLA (Figure 1). The APLA levels were categorised as ≤1 = none, 2 = low, 3 = medium, and 4 = high, according to a template supplied by LIA’s manufacturer.

The peak levels achieved for each COVID-19 patient or HC were plotted in Figure 2. Most APLA showed some degree of reactivity in many of the COVID-19 inpatients and, to some extent, in HC. APLA positivity was evaluated considering the cut-off point established by the 30 HC. Of the eighteen tested APLA, those that did not reach positivity were discarded, namely aAnV IgG, aPG IgG, aPI IgG, aPS IgG, and aPT IgG (Figure 2).

Among the APLA with positivity, aCL IgM presented the highest prevalence (13.04%; 95% CI: 7.21 to 22.06), followed by aPG IgM (8.70%; 95% CI: 4.10 to 16.90) (Table 2). In our cohort, we observed a prevalence of any APLA positivity of 32.61% (95% CI: 23.41 to 43.28). A total of 10 patients (10.87%; 95% CI: 5.62 to 19.51) were positive for more than one type of APLA antibody. The demographic and pre-clinical characteristics of Table 1 were compared between patients who tested positive or negative for any APLA (Table 1). APLA positivity was evenly distributed by sex (*p* = 0.51) and age (*p* = 0.26). No differences were observed in other comorbidities between the APLA-positive and APLA-negative patients (Table 1).

### 3.2. APLA Is Associated with COVID-19 Thrombosis and Inflammation

We performed univariate analyses to look at associations between APLA and COVID-19-related features (Table 3). Patients with positive aCL IgM had a more frequent history of thrombosis than patients with negative aCL IgM (16.7% vs. 1.2%, *p* = 0.044). However, positive aCL IgM patients showed the same frequency of COVID-19-related thrombosis as patients with negative aCL IgM (*p* = 0.23). Of note, patients with positive aPS IgM showed a higher occurrence of COVID-19-related thrombosis than patients with negative aPS IgM (40.0% vs. 5.8%, *p* = 0.046). In addition, patients with positive aPI IgM had higher levels of IL-6 and ferritin than those with negative aPI IgM (150 [88.2, 335] pg/mL vs. 27.5 [10.6, 63.5] pg/mL, *p* = 0.007; 1657 [630, 2366] ng/mL vs. 472 [259, 810] ng/mL, *p* = 0.034; respectively). Finally, those patients with aPE IgG positivity had a longer prothrombin time (expressed as the international normalised ratio, PT/INR) than patients with negative aPE IgG (1.4 [1.3, 1.4] vs. 1.1 [1, 1.2]; *p* = 0.041).

Nevertheless, none of the positive APLAs were associated with COVID-19 severity in a univariate analysis (Table 4). The risk for COVID-19 severity was calculated by logistic regression for aCL IgM (OR [95% CI] 1.13 [0.16, 5.02], *p* = 0.88), aPS IgM (1.42 [0.07, 10.62], *p* = 0.76), and aPI IgM (4.17 [0.51–27.78], *p* = 0.14). These figures revealed that positivity for these APLAs was not a risk factor for COVID-19 severity.

### 3.3. Laboratory Data on Admission and Disease Outcome between APLA Positive and Negative COVID-19 Inpatients

We analysed whether any APLA positivity was associated with some laboratory parameter on admission and at 48 h of hospitalisation (Figure 3 and Appendix A). APLA-positive and APLA-negative patients who had lymphocytopenia presented higher levels of LDH activity, D-dimer, ferritin, and CRP at admission and 48 h later, and IL-6 was higher on admission than the reference values. Nevertheless, no differences were observed on these laboratory parameters between APLA-positive and APLA-negative patients. Regarding clinical outputs, APLA positivity was not associated with longer stays in the hospital, COVID-19 severity, or mortality (Table 5). In regard to anticoagulant treatment, it was not differently administered between APLA-positive and APLA-negative patients (Table 5).

### 3.4. APLA Positivity by Sex in COVID-19 Hospitalised Patients

We looked into differences by sex, and we observed APLA positivity in men but not in women correlated with higher levels of ferritin on admission (median and 25th–75th percentile of APLA-negative men vs. APLA-positive men: 632 [364, 927] vs. 1021 [630, 2209] ng/mL, *p* = 0.006; APLA-positive men vs. APLA-positive women: 1021 [630, 2209] vs. 257 [148, 340] ng/mL, *p* = 0.001), and were kept at 48 h of hospitalisation (APLA-negative men vs. APLA-positive men: 822 [422, 1360] vs. 1192 [528, 2442] ng/mL, *p* = 0.042; APLA-positive men vs. APLA-positive women: 1192 [528, 2442] vs. 260 [200, 368] ng/mL, *p* = 0.006) (Figure 4A). Although procalcitonin, an indicator of bacterial infection or superinfection, remained on average at physiological levels, APLA-positive men showed higher levels than those who were APLA-negative (0.20 [0.18, 0.33] vs. 0.11 [0.06, 0.14] ng/mL, respectively, *p* = 0.009) and APLA-positive women (0.20 [0.18, 0.33] vs. 0.13 [0.03, 0.14] ng/mL, respectively, *p* = 0.009) (Figure 4B). No differences were observed in other laboratory parameters associated with APLA positivity. Despite no differences in days of hospitalisation being found between men and women (16.5 [12.2, 24.8] vs. 18.5 [14.0, 46.2] days), mortality was highly associated with men (*p* = 0.007).

### 3.5. APLA as Acute Markers of Long COVID-19

People who have had a SARS-CoV-2 infection may have long-lasting symptoms that improve over time, but others may worsen. The new syndrome called long COVID-19 includes a wide range of symptoms associated with viral chronic fatigue. Microvascular thrombosis and endotheliopathies have also been described in severe cases of long COVID-19 [43]. We wondered if APLA positivity was related to long COVID-19. At the time of discharge, eight of the ninety-two inpatients had died of COVID-19. Of the 84 survivors of the acute phase of COVID-19, we assessed the diagnosis of long COVID-19 after 18 months of follow-up. A total of three of the eighty-four (3.6%) patients died during this period, and seven (8.33%) presented symptoms related to long COVID-19, of which four (57.4%) were APLA positive. However, we did not observe any association of long COVID-19 with positive APLA (Table 4) or with any APLA positivity (*p* = 0.30). APLA positivity was not a risk factor for long COVID-19 (3.0 [0.39, 16.78], *p* = 0.23).

## 4. Discussion

Our results showed a 32.61% prevalence of APLA positivity in patients hospitalised for COVID-19 during the first pandemic outbreak in Catalonia. A total of 10.87% of patients showed positivity for more than one APLA. The LIA test included criteria and non-Sapporo criteria for APLA. Among them, the criteria aCL IgM showed the highest prevalence (13.04%) of APLA positivity, followed by the non-criteria aPG IgM (8.70%), aPI IgM (5.43%), and aPS IgM (5.43%). Patients with positivity for aPI IgM had higher acute inflammatory marker IL-6 levels, and those with positivity for aPS IgM showed a higher occurrence of thrombosis during their hospitalisation for COVID-19. Nevertheless, none of the positive APLAs were related to the severity of COVID-19 or long COVID-19.

Thrombosis is a common complication in severely ill COVID-19 patients, as proinflammatory and procoagulant activation pathways coexist [1,4,7,30,44]. The prior presence of APLA or “de novo” APLA positivity could play a role in the pathogenesis of these thrombotic events [18,23,24,45]. Some APLA antibodies have been detected in patients with COVID-19, but their relationship with thrombosis and mortality is still a matter of debate [23,37,46,47,48]. We also found that IgM aCL positivity was the most prevalent and, despite being related to previous thrombosis, was not related to current thrombosis disease. Overall, positive APLA were not associated with the severity of COVID-19 or long COVID-19, as was not the case for any APLA positivity. In our cohort, the laboratory parameters associated with the severity of COVID-19 were a lower number of platelets or lymphocytes, a higher number of neutrophils, and elevated levels of ferritin or IL-6. Although these last two parameters were associated with aPI IgM positivity, this was not related to COVID-19 severity. The clinical interpretation of hyperferritinaemia is complex since it may be the result of increased release of ferritin in response to stimuli such as cytokines, oxidants, and hypoxia [49]. Therefore, it can be considered a non-specific marker of pathology without knowing whether concomitant hyperferritinaemia has a causal role in COVID-19 disease.

The association between various infectious agents, mainly bacteria and viruses, and the induction of APLA antibodies has long been recognised [27,50,51]. HIV, varicella-zoster, hepatitis C virus, and many others have been strongly related to APLA positivity of the IgG or IgM isotype, depending on virus class [52]. Most of these virus-induced APLA antibodies were against the lipid-binding domain of proteins such as β2GPI [53]. It should come as no surprise that SARS-CoV-2 is capable of inducing autoantibodies, including APLA, although their persistent expression is uncertain [23,54]. The APLA thrombogenic activity is also undisclosed, as APLA induced by SARS-CoV-2 did not recognise domain 1 of β2GPI protein and could be different from those present in antiphospholipid syndrome [55], as it was previously indicated in other infections [56]. In our cohort, COVID-19-related thrombotic and inflammatory events were associated with aPS and aPI, respectively, and both were of the IgM isotype. Although the clinical implications of transient induction of APLA by viruses have yet to be fully defined, a review of 163 published cases of virus-associated APLA antibodies found thrombotic events in 116 cases [52]. Moreover, the presence of prothrombotic autoantibodies was detected in the serum of patients with COVID-19 [30]. In these studies, APLA prevalence was around 30% [30,35]. Although we used a different methodology, we observed a similar prevalence (32.61%; 95% CI [23.41 to 43.28]).

Critically ill APLA-positive COVID-19 patients had higher median concentrations of CRP and D-dimer and were more likely to have a critical clinical course and fatal outcome [57]. Our cohort showed elevated average levels of CRP and D-dimer. However, these dysregulated levels were not associated with either APLA positivity or COVID-19 severity. The lack of association between APLA positivity and worse clinical outcomes in patients with COVID-19 was also noted by Espinosa et al. [35], and they noted that thrombosis was not related to aCL positivity. Our study is in accord with Espinosa et al. [35], who found that APLA positivity was neither correlated with severe respiratory failure nor mortality.

Among the APLA, special attention is paid to aCL and aβ2GPI, as these APLA are known to be associated with APS, an autoimmune disorder that can lead to blood clots. A meta-analysis [38] found that critically ill patients with COVID-19 had a higher prevalence of aCL and aβ2GPI than non-critically ill patients. However, there was no association between aCL or aβ2GPI positivity and disease outcome, a similar result observed in our study.

Interestingly, we observed differences by sex. Our data indicate that men with elevated ferritin or procalcitonin levels at admission are significantly more likely to test positive for APLA than those with lower levels of these markers. Differences in immunological responses to SARS-CoV-2 infection by sex were reported [58,59,60]. A Tukey’s post-hoc test revealed significant pairwise differences between APLA-positive and APLA-negative patients (+574.17 ng/mL in the APLA-positive group) and between women and men (+735.19 ng/mL in the men). We discussed above that hyperferritenemia could result from cytokine stimuli. Additionally, our data show that IL-6 and CRP were higher in men than women, along with more severe lymphopenia in men, but independent of APLA positivity. Overall, we can guess that the higher mean levels of ferritin (five times above the limit of physiological levels) in men than in women, together with higher levels of inflammatory markers, could stimulate the production of APLA.

Since the first report by Zhang et al. [29] of severely ill patients with COVID-19 complicated by stroke, successive case reports, reviews, and meta-analysis on the association between COVID-19 and APLA have been reported [23,36,37,38]. In most of them, the APLA test was performed once, making it difficult to know whether the APLA were transiently or persistently positive, and the reported APLA positivity was for lupus anticoagulant (LA). In our cohort, 65 (70.1%) patients were tested more than once. Nevertheless, the blood samples were drawn close together, less than 12 weeks apart. Unfortunately, no serum samples were collected during the 18 month follow-up. This is a limitation of the study to verify the persistence of APLA. Another limitation could be that we did not test for LA. However, the LA test could be affected by high CRP levels and anticoagulant therapy, leading to false positive tests [61,62]. This is why we did not include LA testing in our cohort study because patients underwent anticoagulant therapy and had median high CRP levels on admission (median [IQR] positive APLA 107.4 [85.2] mg/L vs. negative APLA 74.9 [130.4] mg/L, *p* = 0.37). Even in those patients with a true positive LA, it will indicate interference in the common blood coagulation pathway, but without identifying which APLA is responsible for it. The multiplex LIA methodology used in our study evaluated 18 different APLAs and identified unique APLAs whose levels are high in serum patients [40,63]. Our finding that only aPS IgM is associated with thrombosis is consistent with the previous description that aPS positivity was associated with higher thrombotic risk in LA-positive patients [64]. It could be that the presence of these autoantibodies and thrombotic events is random since it is difficult to know if APLA or other disease-related conditions are the real cause of thrombosis in critically ill patients. For instance, platelets are key regulators of thrombosis and inflammation and are good candidates for mediating COVID-19-associated pathogenesis [65]. Consistent with this, a multivariate logistic regression analysis of our cohort pointed to platelet count as a risk factor for COVID-19 severity (OR = 0.38, 95% CI [0.14–0.84], *p* = 0.036). In addition, a link between coagulopathy and SARS-CoV-2-induced molecular changes in infected platelets has been reported [66].

Although thrombosis was associated with poor prognosis [7], we did not observe aPS IgM positivity as a risk factor for COVID-19 severity (1.42 [0.07–10.62], *p* = 0.76). Furthermore, our data analysis did not show any association between thrombosis and disease severity (*p* = 0.07). On the other hand, the logistic regression analysis of our data showed IL-6 on admission as a risk factor for COVID-19 severity (OR = 3.52, 95% CI [1.71–8.67, *p* = 0.002), and higher IL-6 levels were associated with aPI IgM positivity. However, this APLA was also not a risk factor for disease severity (OR = 4.17, 95% CI [0.51–27.78], *p* = 0.14).

Persistence of various symptoms in patients who have recovered from COVID-19, defined as long COVID-19 or post-COVID-19 syndrome, was initially reported with a prevalence of 76% [67], decreasing to 0.43% in a meta-analysis on global data [68]. To date, biological markers to predict long COVID-19 are not yet well established, and one case report associated it with persistence of aCL IgG [69]. In our cohort, the 18 month follow-up reported seven cases of long COVID-19 from eighty-one COVID-19 survivors (the prevalence of long COVID-19 was 8.6%). Of the seven long COVID-19 haulers, four tested APLA-positive in the acute phase of COVID-19. Our data found an association of long COVID-19 with COVID-19 severity but did not find an association of long COVID-19 with APLA positivity in the acute phase of the disease.

Finally, we briefly recall the limitations of our study, already presented and discussed above, which are that we did not test for LA and did not test for APLA positivity twice with a separation greater than twelve weeks.

## 5. Conclusions

In conclusion, we examined positivity for 18 criteria and non-criteria APLA with a LIA multiplex in a cohort of 92 COVID-19 patients during the first wave of the pandemic, with a prevalence of APLA positivity of 32%. Two non-criteria APLA, aPS IgM, and aPI IgM were associated with COVID-19-related thrombosis and acute hyperinflammation, respectively. In addition, aCL IgM positivity was associated with previous thrombosis. We would recommend testing for non-criteria aPL to assess the risk of thrombosis in hospitalised COVID-19 patients. However, APLA positivity was not a risk factor for COVID-19 severity or long COVID-19.

## Figures and Tables

**Figure 1 biomedicines-11-02301-f001:**
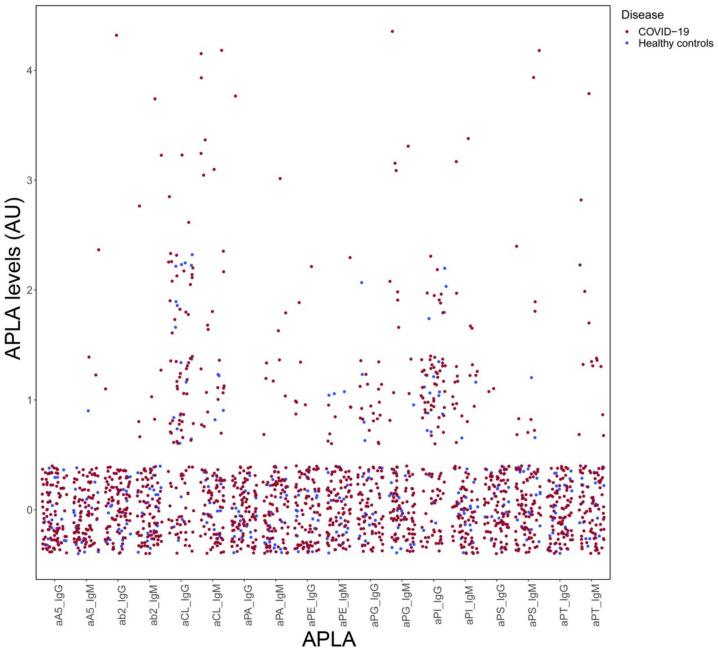
APLA levels in COVID-19 inpatients. Dot plot of levels of APLA in 30 serum samples from HC (blue dots) and 298 serum samples from 92 COVID-19 patients (red dots). A cut-off for each APLA was established based on maximum APLA levels of HC: ≥2 for aA5 IgG, aA5 IgM, ab2 IgG, ab2 IgM, aCL IgM, aPA IgG, aPA IgM, aPE IgG, aPE IgM, aPG IgM, aPI IgM, aPS IgG, aPS IgM, aPT IgG, and aPT IgM; and ≥3 for aCL IgG, aPG IgG, and aPI IgG. APLA that did not reach positivity for any serum sample were aA5 IgG, aPG IgG, aPI IgG, aPS IgG, and aPT IgG. Abbreviations for APLA are: aA5, anti-annexin V; ab2, anti-β2 Glycoprotein I; aCL, anti-cardiolipin; aPA, anti-phosphatidic acid; aPE, anti-phosphatidylethanolamine; aPG, anti-phosphatidylglycerol; aPI, anti-phosphatidylinositol; aPS, anti-phosphatidylserine; aPT, anti-prothrombin; Ig, Immunoglobulin.

**Figure 2 biomedicines-11-02301-f002:**
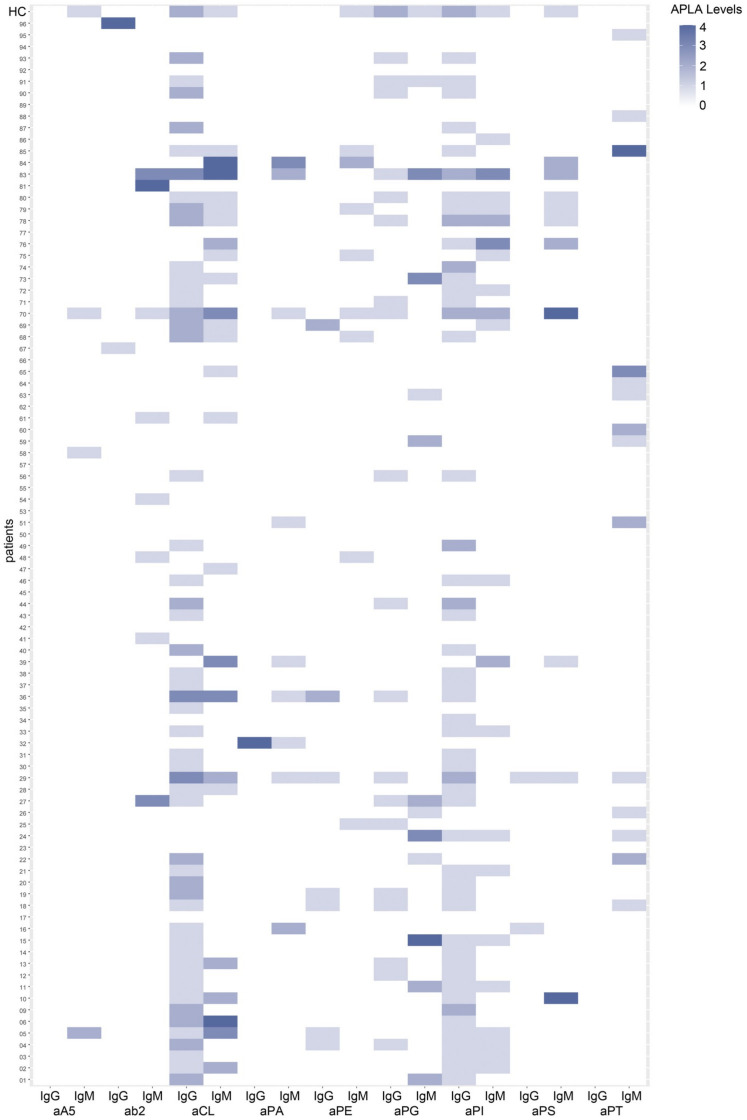
APLA levels in serum samples from 92 hospitalised COVID-19 patients. Heatmap presentation of APLA levels from 92 hospitalised COVID-19 patients and 30 healthy controls tested by LIA for 18 different APLA antibodies. APLA levels were assessed according to the template provided by the LIA manufacturer (GA GmBH): ≤1 = none, 2 = low, 3 = medium, and 4 = high. COVID-19 inpatients on the *y*-axis had multiple blood draws on different days during their hospital stay, and their peak APLA levels are plotted, including the peak APLA levels of the 30 healthy controls (HC on the top row of the heatmap). The APLA positivity was considered when the APLA level achieved by the COVID-19 patient was above the maximum APLA level achieved by the sera of 30 healthy controls. Thus, aCL IgG, aPG IgG, and aPI IgG were considered positive with levels ≥ 3, and for the rest of APLA with levels ≥ 2. Five APLAs did not reach APLA positivity: aA5 IgG, aPG IgG, aPI IgG, aPS IgG, and aPT IgG. Abbreviations for APLA are: aA5, anti-annexin V; ab2, anti-β2 Glycoprotein I; aCL, anti-cardiolipin; aPA, anti-phosphatidic acid; aPE, anti-phosphatidylethanolamine; aPG, anti-phosphatidylglycerol; aPI, anti-phosphatidylinositol; aPS, anti-phosphatidylserine; aPT, anti-prothrombin; Ig, Immunoglobulin Schemes follow the same formatting.

**Figure 3 biomedicines-11-02301-f003:**
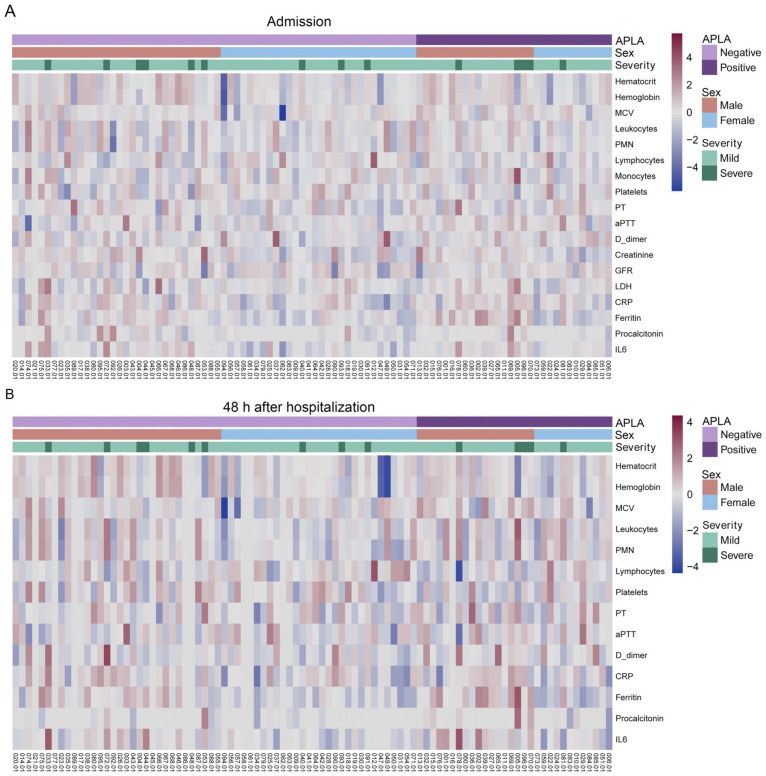
Laboratory data on admission for COVID-19 patients. Heatmap of laboratory data from COVID-19 inpatients on admission (**A**) and at 48 h of hospitalisation (**B**). Red, blue colour depict z-scores. Patients with a positive or negative APLA test were grouped as indicated in the top banner. Subsequently, they were grouped by sex (middle banner). Finally, the severity of COVID-19 was indicated in the bottom banner. Laboratory data values are shown in logarithmic form. No clustering of any laboratory parameter was observed except for ferritin, which clusters depending on APLA positivity and sex (see Figure 4A).

**Figure 4 biomedicines-11-02301-f004:**
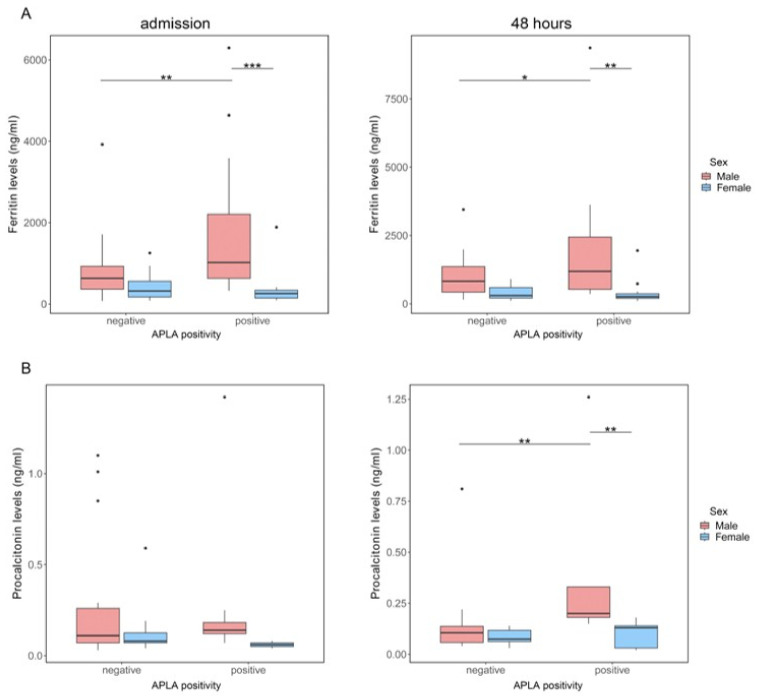
Blood markers associated with APLA positivity. Levels of ferritin (**A**) or procalcitonin (**B**) on admission (**left column**) or at 48 h of hospitalisation (**right column**) of patients who were grouped by whether they had any positive APLA and by their sex (male, red; female, blue). Two-way ANOVA with Tukey’s post-hoc analysis. * *p* < 0.05; ** *p* < 0.01; *** *p* < 0.001.

**Table 1 biomedicines-11-02301-t001:** Demographic and pre-clinical data of COVID-19 hospitalised patients.

	Total (N = 92)	APLA Positive (N = 30)	APLA Negative (N = 62)	*p*-Value
Sex (female) ^1^	42 (45.7%)	12 (40.0%)	30 (48.4%)	0.51 †
Age (years) ^2^	63.67 (13.0)	65.8 (11.9)	62.6 (13.5)	0.26 ‡
Hypertension ^1^	34 (37.0%)	15 (50.0%)	19 (30.6%)	0.11 †
Dyslipidaemia ^1^	29 (31.6%)	8 (26.7%)	21 (33.9%)	0.63 †
Renal chronic disease ^1^	11 (12.0%)	3 (10.0%)	8 (12.9%)	1.00 †
Diabetes Mellitus ^1^	18 (19.6%)	6 (20.0%)	12 (19.3%)	1.00 †
Myocardiopathy ^1^	8 (8.7%)	2 (6.7%)	6 (9.7%)	1.00 †
Peripheral vasculopathy ^1^	5 (5.4%)	2 (6.7%)	3 (4.8%)	0.66 †
Stroke ^1^	4 (4.3%)	0 (0.0%)	4 (6.4%)	0.30 †
COPD-Asthma ^1^	10 (10.9%)	2 (6.6%)	8 (12.9%)	0.49 †
ILD ^1^	1 (1.1%)	0 (0.0%)	1 (1.6%)	1.00 †
AID ^1^	8 (8.7%)	4 (13.3%)	4 (6.4%)	0.43 †
Thrombotic history ^1^	3 (3.3%)	2 (6.7%)	1(1.6%)	0.25 †
Inherited thrombophilia ^1^	1 (1.1%)	1 (3.3%)	0 (0.0%)	0.33 †
Cancer ^1^	10 (10.9%)	3 (10.0%)	7 (11.3%)	1.00 †

^1^ N (%); ^2^ mean (standard deviation, SD). COPD-Asthma—chronic obstructive pulmonary disease; ILD—interstitial lung disease; AID—autoimmune diseases. † Fisher’s exact test; ‡ Mann–Whitney U test.

**Table 2 biomedicines-11-02301-t002:** Number of COVID-19 inpatients with APLA positivity and its average prevalence with 95% confidence interval (CI).

APLA ^1^	n	Positives	Prevalence (%)	95% CI
aCL IgG	92	3	3.26	0.85–9.91
aPA IgG	92	1	1.09	0.06–6.76
aPE IgG	92	2	2.17	0.38–8.38
aβ2GPI IgG	92	1	1.09	0.06–6.76
aCL IgM	92	12	13.04	7.21–22.06
aPA IgM	92	3	3.26	0.85–9.91
aPE IgM	92	1	1.09	0.06–6.76
aPG IgM	92	8	8.7	4.1–16.9
aPI IgM	92	5	5.43	2.02–12.81
aPS IgM	92	5	5.43	2.02–12.81
aAn5 IgM	92	1	1.09	0.06–6.76
aβ2GPI IgM	92	3	3.26	0.85–9.91
aPT IgM	92	5	5.43	2.02–12.81

^1^ Only those APLAs that tested positive are presented here. aAn5 IgG, aPG IgG, aPI IgG, aPS IgG, and aPT IgG did not reach positivity as shown in Figure 2.

**Table 3 biomedicines-11-02301-t003:** Significant clinical and laboratory data from COVID-19 inpatients on admission associated with APLA positivity.

APLA	Parameter	N ^1^	APLA Negative	APLA Positive	*p*-Value	Adjusted *p*-Value
aCL IgM	Thrombotic history †	92	1; 1.2% [0, 6.8]	2; 16.7% [2.1, 48.4]	0.044	0.046
aPS IgM	COVID-19-related thrombosis †	92	5; 5.8% [1.9, 13]	2; 40% [5.3, 85.3]	0.046	0.050
aPI IgM	IL-6 ††	72	27.5 [10.6, 63.5]	150 [88.2, 335]	0.007	0.012
aPI IgM	ferritin ††	82	472 [259, 810]	1657 [630, 2366]	0.034	0.041
aPE IgG	INR ††	78	1.1 [1.0, 1.2]	1.4 [1.35, 1.45]	0.041	0.043

Only those APLAs with a significant association with a clinical or laboratory parameter are presented. † For categorical variables (thrombotic history and COVID-19-related thrombosis), it shows the number of people presenting the clinical parameter distributed in whether they were positive or negative for the indicated APLA; percentage results from dividing it by the number of people testing positive or negative for that aPL according to Table 2 (n; % [95% CI], Fisher’s exact test). †† For numerical variables (IL-6, ferritin, and PT), it shows the median value and IQR of that clinical parameter for those patients who tested positive or negative for the indicated APLA (median [IQR] and Mann–Whitney U test). ^1^ The n column indicates the total number of patients who were analysed for the indicated clinical parameter. IL-6, interleukin-6; INR, international normalised ratio.

**Table 4 biomedicines-11-02301-t004:** Association of APLA positivity with COVID-19 severity or long COVID-19.

	COVID-19 Severity *n* = 92			Long COVID-19 *n* = 81		
	Moderate (*n* = 78) n; %; [95% CI]	Severe (*n* = 14) n; %; [95% CI]	*p*-Value †	No (*n* = 74) n; %; [95% CI]	Yes (*n* = 7) n; %; [95% CI]	*p*-Value †
aCL IgG	3; 3.8; [0.8, 10.8]	0; 0; [0, 23.2]	1.00	2; 2.7%; [0.4, 10.2]	0; 0%; [0, 41]	1.00
aPA IgG	1; 1.3; [0, 6.9]	0; 0; [0, 23.2]	1.00	1; 1.4%; [0, 7.9]	0; 0%; [0, 41]	1.00
aPE IgG	2; 2.6; [0.3, 9]	0; 0; [0, 23.2]	1.00	1; 1.4%; [0, 7.9]	0; 0%; [0, 41]	1.00
aβ2GPI IgG	0; 0; [0, 4.6]	1; 7.1; [0.2, 33.9]	0.15	0; 0%; [0, 5.3]	1; 14.3%; [0.4, 57.9]	0.09
aCL IgM	10; 12.8; [6.3, 22.3]	2; 14.3; [1.8, 42.8]	1.00	8; 10.8%; [5.2, 21.9]	2; 28.6%; [3.7, 71]	0.23
aPA IgM	3; 3.8; [0.8, 10.8]	0; 0; [0, 23.2]	1.00	3; 4.1%; [0.9, 12.4]	0; 0%; [0, 41]	1.00
aPE IgM	1; 1.3; [0, 6.9]	0; 0; [0, 23.2]	1.00	1; 1.4%; [0, 7.9]	0; 0%; [0, 41]	1.00
aPG IgM	8; 10.3; [4.5, 19.2]	0; 0; [0, 23.2]	0.60	7; 9.5%; [4.2, 20.1]	1; 14.3%; [0.4, 57.9]	0.56
aPI IgM	3; 3.8; [0.8, 10.8]	2; 14.3; [1.8, 42.8]	0.17	3; 4.1%; [0.9, 12.4]	0; 0%; [0, 41]	1.00
aPS IgM	4; 5.1; [1.4, 12.6]	1; 7.1; [0.2, 33.9]	0.57	4; 5.4%; [1.6, 14.4]	0; 0%; [0, 41]	1.00
aAnV IgM	0; 0; [0, 4.6]	1; 7.1; [0.2, 33.9]	0.15	0; 0%; [0, 5.3]	1; 14.3%; [0.4, 57.9]	0.09
aβ2GPI IgM	2; 2.6; [0.3, 9]	1; 7.1; [0.2, 33.9]	0.39	3; 4.1%; [0.9, 12.4]	0; 0%; [0, 41]	1.00
aPT IgM	5; 6.4; [2.1, 14.3]	0; 0; [0, 23.2]	1.00	5; 6.8%; [2.4, 16.3]	0; 0%; [0, 41]	1.00

† Fisher’s exact test.

**Table 5 biomedicines-11-02301-t005:** Clinical outputs of COVID-19 disease related to any APLA positivity.

	APLA Positive (N = 30)	APLA Negative (N = 62)	*p*-Value
Days in hospital ^1^	19.0 (28.5)	15.5 (18.8)	0.05 ‡
Severity	5 (16.7)	9 (14.5)	0.77 †
Mortality	2 (6.7)	6 (9.7)	1.00 †
Thrombosis	2 (6.7)	5 (8.2)	1.00 †
LMWH treatment	27 (90.0)	49 (79.0)	0.37 †

N (%); ^1^ mean (SD). ‡ Mann–Whitney U test; † Fisher’s exact test; LMWH low molecular weight heparin.

## Data Availability

Research data generated during this study are saved in our institutional repository at the Vall d’Hebron Institut de Recerca and will be available upon reasonable request to the corresponding authors.

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
