# Peer review of "Thrombosis and Hyperinflammation in COVID-19 Acute Phase Are Related to Anti-Phosphatidylserine and Anti-Phosphatidylinositol Antibody Positivity"

_biomedicines, 2023, doi:10.3390/biomedicines11082301_

Round 1

Reviewer 1 Report

This is a potentially interesting article on APLA antibodies in COVID-19 patients. However, several problems have to be addressed to reconsider the final recommendation, since some concerns with limitations and presentation of the results have been noticed.

1. Please delete words: (1) objective, (2) patients/methods, (3) results, (4) conclusions, from the abstract.

2. Please use APLA as a short form of antiphospholipid antibodies throughout the whole manuscript.

3. Why were healthy donors used to set up the cutoff points for APLA? In several cases, there are cutoff points suggested by producers.

4. Please explain these results: (32.61%; 95% CI [23.41-43.29]), (33,3%; [17.94-52.86]), instead should be: 30 (32.6%), 10 (33.3%)?

5. Values of p<0.05 cannot be accepted, please indicate the exact p-value in each case.

6. Anti-cardiolipin IgM was the most prevalent (13%) - 13% of what?

7. was associated with a history of thrombosis - how?

8. with a prevalence of 5.4% - in the whole group or where, COVID-19 patients?

9. was associated with higher levels of interleukin-6 - positively/negatively? 

10. There are 7 cited papers (>10% of all cited papers in the manuscript) in the first sentence of the Introduction section, why? I recommend limiting it.

11. Lines 42-43, please develop this sentence by introducing possible mechanisms.

12. Why there is no information about lupus anticoagulant (LA)? Only in the discussion section. However, the assessment of LA is relevant and should be done in my opinion.

13. How was the control group recruited? Please add relevant information.

14. Please change: gender into sex, also see Figure.

15. Please round p-value > 0.05 to two decimal places, if p<0.05 then to three decimal places throughout the manuscript also in Supplementary Table S1.

16. I do not understand the 95% CI for the prevalence of APLA. Please clarify if it is necessary.

17. How many patients had previous thrombosis, COVID-19-related thrombosis? What are the definitions of these states? Speculating about the impact of APLA having 3 patients with a history of thrombosis in the COVID-19 group is controversial.

18. Table 3 is confusing and hard to understand. Please clarify. Adjusted p-values, adjusted for what?

19. p-val=0.7661 => p = 0.77, p = 1 => p = 1.00; and next p-values following this rule

20. Lines 214-220, these sentences are hard to understand. The results should be better presented.

21. The conclusion should include information important for clinicians and practitioners. Please add the relevance of these results to daily routine.

22. Why are 95% CI used in Tables 2, 3 and 4? I think that n (number) with % is enough to present such data. Please explain.

23. Table 5. Thrombosis? Which cases of this state are presented? COVID severity? Which one? 

24. The results described in 3.4 should be discussed in the Discussion section.

25. Line 315, noted by Espinosa et al. [38]

26. Lines 317-319. It was shown that specific APLA+ men patients had higher levels of ferritin and procalcitonin. Please rewrite. I cannot see information about creatinine, please add it in the Results section. How could you explain these results? Please add it to the Discussion.

27. Line 324. by Zhang et al. [18]

28. The authors should discuss aCL and aβ2GPI antibodies according to the current literature in the context of COVID-19 since these antibodies are the most recognizable ones. Therefore, the following paper might be interesting in discussing: 10.1016/j.thromres.2023.01.016.

29. There are some limitations in the text. Please add at the end of the Discussion section a separate paragraph titled Limitations, and put all these sentences there.

English editing is required to change style, typos and grammar.

Author Response

Please find in the attachment the point by point author's response to all your comments

Author Response

We uploaded in the attachment the point by point author's response

Round 2

Reviewer 1 Report

I would like to thank the Authors for the changes made, but unfortunately, not all of them have been thoroughly explained. I still have some doubts, so I encourage the authors once again to revise the manuscript in terms of the suggestions below.

1. All changes are accepted, however, please highlight new changes in the manuscript just in red (not in Word using change detection, it is hard to read).

2. Abstract:

I suggest adding 2-3 sentences in the beginning to introduce the topic, it is not good to put the aim of the study in the beginning without any description.

This was done in => Thus, we enrolled

5.4% of inpatients => 5.4% of inpatients (n = 5)

with COVID-19- 25 related thrombosis (pp-value = <0.0465) => 

with the occurrence of COVID-19 related thrombosis (p = 0.046)

(13%) => (n = 12, 13.0%)

with a clinical history of thrombosis (p-value = 27 <0.0445) => with a presence of clinical history of thrombosis (p = 0.044)

of 5.4% => of 5.4% (n = 5)

(pp-value = <0.0071) and ferritin (pp-value = <0.0345) => (p = 0.007) and ferritin (p = 0.034)

In (4)c Conclusion,s: aA third of COVID-19 inpatients tested positive for at least one 32 APLAaPL. Anti-phosphatidylserine IgM associated with thrombosis and anti-phosphatidylinositol 33 IgM associated with inflammation. However, they were not related to disease severity or long 34 COVID. => I suggest some changes, please see and decide:

In conclusion, almost a third of COVID-19 inpatients tested positive for at least one APLA. The presence of anti-phosphatidylserine antibody in IgM class was associated with thrombosis. Moreover, the occurrence of anti-phosphatidylinositol antibody in IgM class was related to inflammation reveled by elevated IL-6 levels. Thus, testing for non-criteria APLA to assess the risk clinical complications of hospitalized COVID-19 patients might be beneficial.

3. I have some concerns about presenting results using 95% CI. Therefore, please let me know if this is a good solution. Otherwise, the decision rests with the editor.

4. Introduction:

SARS‑CoV‑2 => severe acute respiratory syndrome coronavirus 2 (SARS‑CoV‑2)

COVID-19 => coronavirus disease 2019 (COVID-19)

Lines 49-54, citation? What about lupus anticoagulant (LA)?

Lines 55-58, citation?

Lines 62, what about LA, not only aCL and aβ2GPI are criteria APLA.

Line 64, or obstetric failure?, be specific

APLA can also appear related to infections. => please rephrase + citation

Studies looking for the prevalence of aPLAPLA in 71 patients with COVID-19 reported mainly for criteria aPLAPLA, and few delving into non- 72 criteria aPLAPLA, mainly aPS/PT [231–253]. => Please divide citations, firstly cite these articles reporting the prevalence of APLA in COVID-19, doi: 10.1016/j.thromres.2023.01.016, and these articles presenting the occurrence of non-criteria APLA, indeed: Studies reporting the prevalence of APLA in patients with COVID-19 focused mainly for criteria APLA (citations), and only few delving into non-criteria APLA, mainly aPS or aPT (citations).

5. Supplementary Table S1. I believe that this table can be placed in the manuscript since it presents interesting data. Please change aPL into APLA. Please indicate what is in brackets in the table, SD, IQR? Blood cells: should be 109/l; 1.73m2 => 1.73 m2

6. Where there differences in demographic, clinical and laboratory (e.g. APLA) characteristics between COVID-19 patients on anticoagulant treatment and without it? If so, it would be beneficial to add this information and discuss it.

7. I cannot see a table describing a comparison between all COVID-19 patients and controls. Please, provide relevant data including demographic and clinical characteristics (sex, age, BMI, comorbidities). Both groups should be similar according to sex, age, BMI, and possibly with the rest characteristics. Healthy blood donors (HC) - if the control groups suffer from comorbidities, we cannot name them as healthy. Next, in line 161, healthy controls are abbreviated to the form HC.

8. Please add values of body mass index in the Table 1 and relevant comparisons. 

9. Statistics. Line 141-142. Please, indicate when, for normally/non-normally distributed data. Should be: a 95% confidence interval (CI).

10. Table 1. Should be: Sex (female), n (%). Age (years), mean (SD) - and in a footnote: SD - standard deviation. Hypertension, n (%), etc. And for the results: 42 (45.7) => 42 (45.7%), etc. for all without age. COPD-Asthma, chronic obstructive pulmonary disease => COPD-Asthma - chronic obstructive pulmonary disease, etc. for all.

11. Lines 208-209, please change: p-value => p; p-value = 0.5108 => p = 0.51; etc.

12. Lines 209-210, between which groups?

13. Lines 234-242, also 261-262, also 292-304, also 322-323, and discussion. Please change: p-value => p. But, in tables should be: p-value, as it is now.

14. Results of 3.2.

Although aCL IgM, the most prevalent APLA positivity, was associated with previous thrombosis (p-value = 0.044), it was not associated with COVID-19- related thrombosis. => What do you mean by this sentence? This should be changed:

Patients with positive aCL IgM had more frequent previous history of thrombosis comparing with negative aCL IgM patients (16.7% vs. 1.2%, p = 0.044).

Next:

Of note, aPS IgM was associated with thrombosis after 235 SARS-CoV-2 infection (pp-value = <0.0465), while, aPI IgM positivity was associated with 236 higher levels of IL-6 (median [IQR] aPI IgM positive 150 [88.2, 335] pg/ml vs. aPI IgM 237 negative 27.5 [10.6, 63.5] pg/ml, pp-value = <0.0071), and ferritin (aPI IgM positive 1657 238 [630, 2366] ng/ml vs. aPI IgM negative 472 [259, 810] ng/ml, pp-value = <0.0345)

Suggestion:

Of note, patients with positive aPS IgM antibodies were characterized by a higher occurrence of thrombosis after SARS-CoV-2 infection than negative aPS IgM antibodies patients (40.0% vs. 5.8%, p 0.046), while, patients with positive aPI IgM antibodies had increased IL-6 levels than negative aPI IgM (150.0 [88.2-335.0] vs. 27.5 [10.6-63.5], pg/ml, p = 0.007), and ferritin (1657 [630-2366] vs. 472 [259-810], ng/ml, p = 0.034).

Finally, 239 those patients with aPE IgG positivity were correlated with a higher prothrombin time 240 (PT, ratio) relative to those with negative aPE IgG (1.4 [1.3, 1.4] vs. 1.1 [1, 1.2]; pp-value = 241 <0.0415, respectively). =>

Finally, those patients with positive aPE IgG antibodies had prolonged prothrombin time to those with negative aPE IgG antibodies (1.40 [1.35-1.45] vs. 1.10 [1.00-1.20], p = 0.041). = Is it PT or INR? Please specify, and add unit.

15. Table 4. Please change: 0.571 => 0.57. 

16. Results of 3.4. Please change IQR into Q1-Q3

17. Results of 3.5. Where there differences in long-COVID regarding age, sex, comorbidities?

18. Discussion. In the beginning please highlight the occurrence of all APLA, then indicate non-criteria APLA in analyzed COVID-19 patients, with the most prevalent, next, present one sentence regarding differences between on admission/disease outcome in COVID-19, and long-COVID - in one paragraph. The next paragraph should start with: Thrombosis is a common complication in severely ill COVID-19 patients... 

19. Lines 330-331, need more citations, since there are plenty of papers discussing the clinical implications of APLA prevalence in COVID-19, and some of them were previously used.

20. Clinical interpretation of hyperferritinaemia is complex, since it may be the 339 result of increased release of ferritin in response to stimuli such as cytokines, oxidants and 340 hypoxia. - citation?

21. I believe that the paragraph starting in line 343 is too long and should be divided. First paragraph: The association between various infectious agents...; next one: Critically ill APLA...; next: Among the APLA...; and: Interestingly, we observed...; 

22. Line 448. Please add: However, APLA...

Minor editing of English language required. Please check the paper.

Author Response

Please,

find enclosed the authors' responses to your comments

Round 3

Reviewer 1 Report

I would like to congratulate the Authors for their work! I only have a few minor comments that should be taken into account before accepting the article fo publication.

1. Abstract:

Please change: interleukin-6 (IL-6, p = 0.007) => interleukin(IL)-6 (p = 0.007)

33,3% => 33.3%

13% => 13.0%

positivity in IgM class associated => positivity in IgM class was associated

2. Please, write numbers according to citations

Studies reporting the prevalence of APLA in patients with COVID-19 76 focused mainly for criteria APLA (Devresse; Gatto ), and only few delving into non-crite- 77 ria APLA, mainly aPS or aPT [Amezcua; Espinosa).

3. When describing controls, there should be information about the similarity according to at least age and sex between healthy and COVID-19 groups. Please include it (lines 110-116).

4. Line 153. To explore any aPL positivity => APLA

Author Response

We thank the reviewer for the comments and corrections. The reviwer will find our responses in the attached document.
